# Evaluation of a Recombinant Mouse X Pig Chimeric Anti-Porcine DEC205 Antibody Fused with Structural and Nonstructural Peptides of PRRS Virus

**DOI:** 10.3390/vaccines7020043

**Published:** 2019-05-23

**Authors:** Lorena Bustamante-Córdova, Mónica Reséndiz-Sandoval, Jesús Hernández

**Affiliations:** Laboratorio de Inmunología, Centro de Investigación en Alimentación y Desarrollo, A.C., Hermosillo 83304, Sonora, Mexico; lorebcw@gmail.com (L.B.-C.); mresendiz@ciad.mx (M.R.-S.)

**Keywords:** antigen targeting, DEC205, PRRSV, recombinant antibodies, chimeric antibody, dendritic cells

## Abstract

Activation of the immune system using antigen targeting to the dendritic cell receptor DEC205 presents great potential in the field of vaccination. The objective of this work was to evaluate the immunogenicity and protectiveness of a recombinant mouse x pig chimeric antibody fused with peptides of structural and nonstructural proteins of porcine respiratory and reproductive syndrome virus (PRRSV) directed to DEC205^+^ cells. Priming and booster immunizations were performed three weeks apart and administered intradermally in the neck area. All pigs were challenged with PRRSV two weeks after the booster immunization. Immunogenicity was evaluated by assessing the presence of antibodies anti-PRRSV, the response of IFN-γ-producing CD4^+^ cells, and the proliferation of cells. Protection was determined by assessing the viral load in the blood, lungs, and tonsils using qRT-PCR. The results showed that the vaccine exhibited immunogenicity but conferred limited protection. The vaccine group had a lower viral load in the tonsils and a significantly higher production of antibodies anti-PRRSV than the control group (*p* < 0.05); the vaccine group also produced more CD4^+^IFN-γ^+^ cells in response to peptides from the M and Nsp2 proteins. In conclusion, this antigenized recombinant mouse x pig chimeric antibody had immunogenic properties that could be enhanced to improve the level of protection and vaccine efficiency.

## 1. Introduction

Vaccination has been established as the most effective tool for the prevention, control, and eradication of infectious diseases; for this reason, improved vaccines are continuously pursued [1]. The targeting of antigens to dendritic cells (DCs) has been evaluated as a strategy to increase the effectiveness of vaccines. It has been reported that the targeting of C-type lectin receptors (CLRs) in mice and monkeys, even with low concentrations of antigen, is capable of generating a cellular immune response [2]. The targeting of the DEC205 receptor, a member of the family of CLRs, has been evaluated in vivo using anti-DEC205 antibodies conjugated with different antigens, such as ovalbumin, to induce antigen-specific CD4^+^ and CD8^+^ T lymphocyte responses in mice [3].

In murine models, it has been observed that targeting of the DEC205 receptor results in increases in the efficiency of antigen presentation [4], the stimulation of lymphocytes, and the production of IFN-γ [5]. In birds, early and increased production of antibodies was found by targeting antigens from an avian influenza virus strain (H5N2) to the DEC205 receptor [6]. In cattle, a DNA vaccine was used to induce the production in situ of a single chain variable fragment (scFv) directed against the DEC205 receptor, and the scFv was fused to the MSP1a antigen of the tick *Anaplasma marginale* [7]. This approach was successful in inducing a significant response by IFN-γ-producing CD4^+^ cells, the proliferation of CD4^+^ T cells, and increased antibody production, thus showing its immunogenicity.

Recently, the efficiency of targeting the DC-SIGN, Langerin and DEC205 porcine receptors was evaluated using structural proteins of porcine respiratory and reproductive virus (PRRSV) administered intramuscularly [8]. The use of a single chain fragment variable-fragment crystallizable region (scFv-Fc) (mouse x pig) induced a modest but nonsignificant increase in the production of total PRRSV antibodies in the vaccine group targeting DEC205 compared to that in the unvaccinated control group, with no effect on the production in the other target groups (DC-SIGN and Langerin). However, the frequency of IFN-γ-producing CD4^+^ cells was unaltered in the vaccine group targeting DEC205. Ultimately, no decrease in viremia was found, proving a lack of protection. The proteins used in this work included glycoprotein (GP) GP3, GP4, GP5, and the matrix (M) protein. The last two are the major envelope proteins, and these proteins are also considered to be among the most immunogenic proteins and are capable of inducing the production of neutralizing antibodies, especially when used together [9]. Other B cell epitopes have also been found in nonstructural proteins (Nsps), especially Nsp2 [10,11]. Although the humoral response is important in PRRSV infection, cellular mechanisms also contribute to the control of the virus [12]; thus, the stimulation of T cells is a key factor for vaccine effectiveness. Accordingly, other reports have identified several Nsps as having T cell epitopes that have the potential to induce IFNγ production [13,14,15]. To prove the value of Nsps in vaccination, our working group produced a recombinant adenovirus that expressed several peptides from structural proteins and Nsps of PRRSV containing T cell epitopes, and this approach resulted in partial protection in challenged pigs [16]. The mentioned reports raise the possibility of using antigenized recombinant antibodies directed against pig DEC205 to enhance the effectiveness of the immune response and suggest that this approach has potential as a vaccination tool using B and T cell epitopes from PRRSV proteins. As a result, we designed a recombinant chimeric mouse x pig antibody to direct structural and nonstructural peptides of PRRSV to the DEC205 receptor. Here, we evaluated the immunogenicity of the recombinant mouse x pig chimeric antibody and its ability to induce protective immunity against PRRSV in immunized and challenged pigs.

## 2. Materials and Methods

### 2.1. Animals

Six 5-week-old pigs from a PRRSV-free farm were used. Their negative status was confirmed by qRT-PCR and ELISA. The pigs were housed in the facilities of the Centro de Investigación en Alimentación y Desarrollo, A.C. (CIAD, A.C.) with ad libitum access to food and water. Weight gain was monitored weekly throughout the experiment, and temperature changes were monitored during the first week after challenge. The animals were euthanized three weeks after challenge according to the protocols established in the Mexican Official Norm Nom-033-ZOO-1995 for the humane slaughter of domestic animals. The study was approved by the Ethics Committee of CIAD, A.C. (CE/021-B/2014).

### 2.2. Virus Strains and Cell Lines Used

MA104 derived monkey kidney MARC-145 cells were used for the propagation of PRRSV (strain National Veterinary Services Laboratory (NVSL) 97–7895) and were grown in Dulbecco’s modified Eagle medium (DMEM; Gibco) with 10% heat-inactivated fetal bovine serum (FBS; Equinotech) and antibiotics (100 IU penicillin mL^−1^ and 100 μg streptomycin mL^−1^ (Sigma)). After the typical cytopathic effect was observed, cells were collected and lysed by freeze–thaw cycles and centrifuged at 650× *g* at 4 °C for 20 min. Finally, the supernatant was separated, titrated, and stored at −80 °C until further use.

### 2.3. Production of a Recombinant Mouse X Pig Chimeric Antibody

Previously, in our laboratory, a mouse monoclonal antibody (mAb ZH9F7) was produced with classic hybridoma technology using a recombinant protein for the extracellular domains of the DEC205 receptor (GenBank Accession No. GQ420669), as reported by our research group [17]. To construct our chimeric mouse x pig antibody, the mouse Fc domain of the mAb ZH9F7 was replaced with that of a porcine IgG_3_, while the variable light and heavy chains from the original mAb ZH9F7 remained. For the antigenized portion of the chimeric antibody, 12 peptides corresponding to structural and nonstructural proteins of PRRSV were selected from a previous work [16]. The selected peptides as well as their amino acid sequence and genotype (gen) are as follows: nsp2-SV (SLYKLLLEV gen I) [13]; nsp5-IV (ILNEVLPAV gen I) [13]; nsp9-KR (KEEIALSAQIIQACDIR gen II) [15]; nsp10-CL (CPGKNSFLDEAAYCNHL gen II) [15]; Gp5-SL (SHLQLIYNL gen II) [13]; Gp5-CR (CAFAAFVCFVIR gen I) [13]; Gp5-LC (LAALICFVIRLAKNC gen II) [13]; Gp5-KK (KGRLYRWRSPVIVEK gen II) [13]; Gp5-TP (TPLTRVSAEQWGRP gen II) [10]; M-CS (CNDSTAPQKVLLAFS gen II) [14]; M-AL (ALKVSRGRLLGLLHL gen II) [14]; and M-KK (KFITSRCRLCLLGRK gen II) [14]. The coding sequence for the PRRSV peptides was positioned at the carboxy-end of the Fc portion of the chimeric antibody, and each individual PRRSV peptide was separated from the others with two lysine residues (KK). The whole construct for the recombinant mouse x pig chimeric antigenized antibody, named rAb ZH9F7+A1 (Figure 1), was optimized (GenScript) for expression in a eukaryotic system and then cloned into the vector pcDNA 3.1(−) (Invitrogen). Finally, the mammalian cell line Expi293f was transiently transfected to express rAb ZH9F7+A1, following the manufacturer’s protocol (Thermo-Fisher Scientific). The supernatant was collected at three days posttransfection, and the antibody was purified by affinity chromatography using HiTrap Protein A HP 1 mL columns (Sigma-Aldrich).

### 2.4. Pig Immunization and Virus Challenge

The immunization protocol included two groups: (a) the control group (*n* = 3), which was administered 200 μL of sterile phosphate-buffered saline (1X PBS), and (b) the vaccine group (*n* = 3), which was immunized with 150 μg of a chimeric mouse x pig recombinant antibody. This chimeric rAb was antigenized with a chimeric protein consisting of peptides from structural and nonstructural proteins of PRRSV (Figure 1). The immunizations in both groups included 100 μg of adjuvant poly I:C (Polyinosine-Polycytidylic acid, Branched, Sigma-Aldrich). All immunizations were performed intradermally near the neck area. Three weeks after the first immunization, booster doses were administered to both groups in the same manner. Two weeks after the booster immunization, all pigs were challenged with 2 × 10^5^ median tissue culture infectious dose (TCID_50_) of PRRSV intranasally and euthanized three weeks after challenge.

### 2.5. Isolation of PBMCs

Peripheral blood mononuclear cells (PBMCs) were separated from whole blood in the anticoagulant EDTA using Ficoll-Hypaque (GE Healthcare Life Sciences) following the manufacturer’s protocol. After gradient separation, the cells were washed with RPMI-1640 medium (Sigma-Aldrich) supplemented with antibiotics and resuspended in medium containing 10% FBS.

### 2.6. Intracellular IFNγ Detection by Flow Cytometry

Two weeks after challenge, the presence of IFNγ-producing CD4^+^ cells in the PBMC population was evaluated by flow cytometry to assess the T helper lymphocytes 1 (Th1) response to different stimuli. Using 48-well plates, 1 × 10^6^ PBMCs were stimulated for 24 h with 10 μg/mL individual peptides, a multiplicity of infection (MOI) of 0.1 of virus or 20 μg/mL phytohemagglutinin (PHA) as a positive control; unstimulated control wells were included for each pig. The cells were cultured in complete medium (RPMI-1640 medium supplemented with antibiotics, 10% FBS, and β-mercaptoethanol). A protein transport inhibitor cocktail (eBioscience) was added to each individual well 16 h before harvest, as directed by the manufacturer’s instructions. Then, the cells were harvested and extracellular labeling was performed with an anti-CD4 antibody conjugated with fluorescein isothiocyanate (FITC) (Southern Biotech). Afterwards, the cells were fixed and permeabilized with a Leucoperm kit (BioRad) following the maker’s instructions for intracellular labeling with an anti-pig IFNγ antibody conjugated with phycoerythrin (PE) (BD Biosciences). The stimulated cells were acquired and analyzed on a FACS CANTO II™ flow cytometer (BD Biosciences, Franklin Lakes, New Jersey, United States) using FACSDiva software.

### 2.7. CFSE Proliferation Assay

Three weeks after challenge, a proliferation assay was performed using carboxyfluorescein succinimidyl ester (CFSE) staining (BioLegend). The protocol used was a modified version of a previously reported protocol [18]. Briefly, isolated PBMCs were washed with sterile 1X PBS to remove FBS and resuspended in 1 mL of 5 μM CFSE. The cell suspension was incubated for 15 min with gently mixing every 5 min to promote even staining. Next, 1 mL of FBS was added and incubated for an additional minute. Finally, the cells were washed with 10 mL of RPMI-1640 to remove any leftover CFSE and resuspended in complete cell culture medium. Using 48-well plates, 1 × 10^6^ stained cells were stimulated in individual wells with 10 μg/mL individual peptides, an MOI of 0.1 of virus or 10 μg/mL PHA; unstimulated stained and unstained cells were also created for each pig as controls. After four days of incubation, the cells were collected and analyzed using a FACS CANTO II™ flow cytometer (BD Biosciences) and FACSDiva software (BD Biosciences, Franklin Lakes, New Jersey, United States).

### 2.8. ELISA to Detect Specific Anti-PRRSV Antibodies

Serum was collected from all pigs at the beginning of the experiment (Day 0) and on the booster day (Day 21), challenge day (Day 35), and the last day of the experiment (Day 56) to be analyzed for the presence of specific antibodies using a previously established ELISA protocol [16]. Briefly, 96-well plates (Corning) were coated with PRRSV (10^5^ TCID_50_) diluted 1:100 using Carbonate Buffer pH 9.6 overnight at 4 °C. The plates were washed with phosphate-buffered saline and Tween (PBST) (PBS 1X with 0.05% Tween 20; Sigma-Aldrich), blocked with 300 μL of blocking buffer (PBST + 5% fish gelatin; Sigma-Aldrich) per well, and incubated for 1 h at room temperature. After three washes with PBST, 100 μL of each serum sample, at a 1:50 dilution, was added and incubated for 1 h at room temperature. Following three more washes, 100 μL of anti-pig IgG detection antibody conjugated with horseradish peroxidase HRP (1:10,000 dilution) (Southern Biotech Associates, Inc.) was added to each well and incubated at room temperature for 1 h, and the plates were then washed three times. Finally, 50 μL of TMB substrate (Sigma-Aldrich) was added and incubated for 30 min; the plates were then read at 450 nm using a BioRad 680 model microplate reader (Hercules, California, United States). In addition, to confirm the seroconversion of the challenged pigs, serum samples were evaluated with a commercial ELISA kit (Herdchek PPRS Antibody Test Kit, IDEXX Laboratories) following the manufacturer’s protocol. In this case, samples were considered positive if the serum to positive (S/P) values ≥0.4.

### 2.9. PRRSV Detection by qRT-PCR in the Serum, Tonsils, and Lungs

To evaluate viremia in pigs, serum samples were collected at the time of challenge and three weeks postchallenge. Additionally, the presence of PRRSV in the tonsils and lungs at the time of euthanasia was determined. Tissue samples were collected and stored at −20 °C until processed. RNA extraction was performed with tissue samples using the RNeasy Mini Kit (QIAGEN) and with serum using the QIAamp Viral RNA Mini Kit (QIAGEN). SYBR® Green One-Step Real-Time RT-PCR Master Mix (Thermo-Fisher Scientific) was utilized for amplification under the following conditions: 48 °C for 30 min, 95 °C for 10 min, and then 38 cycles of 10 s at 45 °C and 30 s at 60 °C in the StepOne™ Real-Time PCR System (Applied Biosystems, Foster City, California, United States). The primers used to amplify a 198 bp ORF6-7 fragment for genotype 2 PRRSV have been previously reported [19].

### 2.10. Microscopic Pathological Evaluation

Samples of the lungs of all infected pigs were stored in a buffered formaldehyde solution and later evaluated by a pathologist at the Laboratorio de la Asociación de Poricultores (Navojoa, Sonora, México). The lung lesion score assigned to each sample was based on the degree of interstitial pneumonia found according to the following scoring system: 0—no significant lesions, 1—mildly significant lesions, 2—moderately significant lesions, and 3—severely significant lesions, according to the pathologist’s expertise.

### 2.11. Statistical Analyses

GraphPad Prism v6.0 software (San Diego, California, United States) was used to evaluate significant differences between the control and vaccine groups. Two-way ANOVA and Bonferroni multiple comparisons tests were performed for ELISA and viremia analyses. For all other data analyses, Student’s *t*-tests were applied. A *p* ≤ 0.05 was considered statistically significant.

## 3. Results

### 3.1. Clinical Signs

At the beginning of the experiment, the rectal temperature of the pigs in the control group was 39.7 ± 0.3, while the pigs in the vaccine group had a rectal temperature of 39 ± 0.6 °C, which was considered normal for swine. All pigs presented a fever (>40 °C) during the first week after challenge, with peaks on the second and third days at 41.2–41.4 °C in the vaccine and control groups, respectively. The temperature in both groups continued to fluctuate as the days passed, and by day seven postchallenge, the pigs in both groups returned to very similar temperatures below 40 °C.

The pigs in both groups began with similar body weights, namely, 11.6 ± 1.9 kg and 11.6 ± 2.1 kg in the control and vaccine groups, respectively. Nonetheless, the pigs in the control group showed a slightly higher average daily weight gain (ADWG) than those in the vaccine group during the experiment. The control group pigs had an ADWG of 0.44 ± 0.14 kg from the vaccination day to the challenge day and an ADWG of 0.51 ± 0.22 kg from the challenge day to the necropsy day at the end of the experiment. In contrast, the vaccine group had an ADWG of 0.35 ± 0.11 kg from the vaccination day to the challenge day and an ADWG of 0.37 ± 0.17 kg from the challenge day to the necropsy day. However, the difference between the groups was not significant (*p* > 0.05).

### 3.2. Frequency of Specific IFN-γ-Secreting CD4^+^ Cells

To determine the immunogenicity of the recombinant mouse x pig chimeric antibody, the frequency of CD4^+^IFNγ^+^ cells was evaluated ex vivo in response to the peptides contained in the antigenized antibody. Figure 2A shows a representative flow cytometry analysis to determine the presence of CD4^+^IFNγ^+^ cells. The results show an overall low CD4^+^ IFN-γ-secreting cell response to the peptides and virus in both groups. Detectable frequencies of CD4^+^IFNγ^+^ cells in response to the Gp5-KK, GP-LC, and GP5-SL peptides were mainly observed in the control group, while no pigs in the vaccine group responded to these peptides (Figure 2B). Detectable frequencies of CD4^+^IFNγ^+^ cells in the vaccine group occurred mostly in response to M-AL, M-CS, Nsp2-SV, and Nsp-KR. With the exception of the peptide M-AL, the peptides produced a stronger response in the vaccine group than in the control group. Finally, only one pig in each group responded to the virus, albeit with differences in frequency. Nonetheless, there was no statistical significance between the groups for any of the stimuli.

### 3.3. Proliferation of PBMCs in Response to Individual Peptides or Virus

The proliferative response of PBMCs to different stimuli is expressed as a cell percentage (Figure 3). The results showed that at least one pig in the vaccine group had the capacity to proliferate in response to all the peptides and the virus, which was not the case in the control group, where the proliferative response was almost null. All the pigs in the vaccine group responded to the GP5-LC, GP5-SL, M-AL, and M-KK peptides, albeit with differences in magnitude. For the peptides M-CS, Nsp10-CL, and Nsp9-KR and the virus, the vaccine group had two responding pigs, while the control group had a similar response to the M-CS peptide but no response to any of the Nsps peptides or the virus. However, there was no significant difference in the response to any of the stimuli between the groups.

### 3.4. Anti-PRRSV Antibody Response

We compared the antibody response four times, the beginning of the experiment (prime), the booster day, the challenge day, and three weeks postchallenge, by ELISA (Figure 4A). Overall, we found that the vaccine group had a higher antibody response to the virus PRRS than did the control group. Particularly, on the day of the booster immunization (Day 21), on the day of the challenge (Day 35), and at week three postchallenge (Day 56), the antibody levels were significantly greater in the vaccine group than in the control group (*p* ≤ 0.05). The ELISA (IDEXX) showed that, by week three postchallenge, all infected pigs had seroconverted and presented a strong positive status (*p* ≤ 0.001) (Figure 4B).

### 3.5. Viremia and the Viral Load in the Lungs and Tonsils

To evaluate protection in the vaccinated pigs, viremia and viral loads were evaluated. Seven days after challenge, the control and vaccine groups were both viremic, and there was no significant difference between them (Figure 5A). The following week, decreased viremia was observed in both groups, and by three weeks postchallenge, only two pigs in each group had detectable viral levels. Likewise, the viral load in the lungs at the end of the study was not different between the control and vaccine groups, and one pig in each group was negative (Figure 5B). In contrast, the viral load in the tonsils showed that only one pig in the vaccine group was positive, and this pig had a lower viral load than the two positive pigs in the control group (Figure 5C).

### 3.6. Microscopic Lung Lesion Evaluation

We observed similar microscopic lesions indicative of interstitial pneumonia due to PRRSV infection in both groups (Figure 6). In the control group, only two pigs had lesions, one slight and the other moderate, while in the vaccine group, two pigs had moderate lesions. One pig in each group had no apparent pulmonary lesions. The differences between the groups were not statistically significant (Table 1).

## 4. Discussion

Previous reports have demonstrated that antigen targeting to different DC receptors results in increased antibody production, cell activation, and cytokine secretion [20]; thus, antigen targeting using recombinant antibodies has been presented as a potential tool for effective vaccination in veterinary medicine [21]. The persistence of PRRSV in porcine farms over time continues to drive the constant development of vaccines to prevent infection outbreaks and maintain farms in a negative status to reduce the economic impact of PRRSV [22,23]. Overall, the improvement of current vaccination tools and the development of improved vaccination tools are much needed in porciculture around the world; thus, we are attempting to improve current results with an antigen targeting approach to control PRRSV infection. In the present work, we show that antigen targeting of structural and nonstructural peptides of PRRSV towards DEC205^+^ DCs using a recombinant chimeric mouse x pig antibody is immunogenic but produces limited protective immunity in challenged pigs.

The cellular response was evaluated by measuring the frequency of specific CD4^+^ IFN-γ-secreting cells and PBMC proliferation. Interestingly, the peptides that elicited a higher frequency of specific CD4^+^IFN-γ-secreting cells in the vaccine group were part of the M protein (M-AL, M-CS, and M-KK), which has been previously identified as a potent inducer of IFN-γ [16]; in contrast, the control group had a predominant response to peptides from GP5 (GP5-KK, GP5-LC, and GP5-TP), the most variable protein in the virus [16]. However, we did not observe a significant IFN-γ response towards the nonstructural peptides, including those from Nsp10 and Nsp9, even though these proteins have been classified as strong IFN-γ inducers [15]. These findings are in accord with prior reports from our research group [13,24,25], which suggest that the induction of memory T cells against these antigens requires a large number of antigen exposures [24]. Our results are also in agreement with previous results showing that the use of DEC205 targeting of viral proteins (GP5, GP4, and GP3) produces a null CD4^+^ cell response [8]. The use of DEC205 to target dendritic cells has been shown to induce strong IFN-γ production in humans and mice [8,16]. Nevertheless, in our results, the response involving typical Th1 cytokines, such as IFN-γ, was poorly stimulated.

An in-house-created indirect ELISA evaluating the antibody response to the whole virus measured the humoral response to PRRSV. Interestingly, compared to the control group, the vaccine group presented enhanced production of antibodies against PRRS after the first immunization until the end of the experiment. The GP5 and M proteins are known to have B cell epitopes and thus enhance the humoral response [26]. Nonetheless, our results clearly showed an enhanced humoral response compared to the responses measured in other targeting studies where larger antigenic proteins were used [8,27], thus further indicating an improvement in the vaccination strategy used in this work. It is possible that the use of a recombinant chimeric mouse x pig antibody via intradermal application can increase the immunogenicity of this useful strategy. Unfortunately, neutralizing antibodies were not evaluated in this study.

Regardless of the elevated level of anti-PRRSV antibodies detected, we did not observe a difference in viremia or the lung viral load between the vaccine and control groups. However, we found that, in the tonsils, the vaccine group contained two pigs that were negative for PRRSV, while the control group contained two pigs with a high viral load. Surprisingly, the tonsils are considered to be a site of PRRSV persistence, as are the lungs, although this persistence did not appear to occur in our study. Moreover, microscopic lung lesions did not show any significant differences between the groups, which both included pigs with mild to moderate lesions. Similar results for viremia, the lung viral load and microscopic lesions were observed in other studies where structural and nonstructural peptides were used [8,16]. In contrast to our findings, the results of Subramaniam et al. showed that antigen targeting of the structural peptides GP3, GP4, GP5, and M to DEC205^+^ DCs fails to induce a significantly enhanced humoral response or cellular response, as measured by IFN-γ-secreting cells, and has no effect on viremia, the lung viral load or microscopic lung lesions in immunized pigs [8]. The differences between that study and ours are most likely due to the selected immunogenic peptides, the use of a recombinant chimeric mouse x pig antibody and the administration route used, as the intradermal approach is more suitable because of the prevalence of DCs is higher in the skin than in the muscle [28]. Moreover, the injection site for our antigenized rAb was near the submaxillary lymph nodes and tonsils, which have been shown to have a higher percentage of DEC205^+^ DCs than other secondary lymphoid organs, such as the spleen [29]. This increased percentage would mean greater exposure to the targeted cells and lymph node trafficking and could translate into a more efficient vaccine performance [30].

However, as evidenced by the results, not all the peptides selected in this study induced a strong immunogenic response. Replacing the less immunogenic peptides with others that are more reactive or simply repeating the more immunogenic peptides could augment the probability of immunogenic peptide presentation and induce a stronger response. In line with this supposition, since approximately 20% of the protein corresponds to the antigenized region, which means that each peptide accounts for 1.6% of the whole structure, a higher dose of the vaccine could be necessary to produce a greater response and generate superior protective immunity. Thus, higher doses and different antigens need to be evaluated to enhance the immunogenicity of this vaccine. Furthermore, the use of more potent adjuvants could also improve the effectiveness of the vaccine; while poly I:C is considered a potent adjuvant, polyinosinic-polycytidylic acid and poly-L-lysine double-stranded RN (poly ICLC), a derivate of poly-IC stabilized with poly-l-lysine and carboxymethylcellulose, has shown promising results in nonhuman primates [31,32].

## 5. Conclusions

In summary, the results show that antigen targeting of structural and nonstructural peptides of PRRSV towards DEC205^+^ DCs using a recombinant chimeric mouse x pig antibody exhibits immunogenicity but produces limited protective immunity in challenged pigs. It is important to note that inducing protective immunity against PRRSV is a substantial challenge. For this reason, several adjustments, including replacing the antigens that showed the lowest immunogenicity with others that are more immunogenic, could be made, and using a higher dose of the vaccine could definitely increase protection. However, it is important to mention that the immunogenicity observed with the use of the chimeric mouse x pig anti-DEC205 antibody can be used in other infection models. Further studies are needed to probe the application of this novel strategy.

## Figures and Tables

**Figure 1 vaccines-07-00043-f001:**
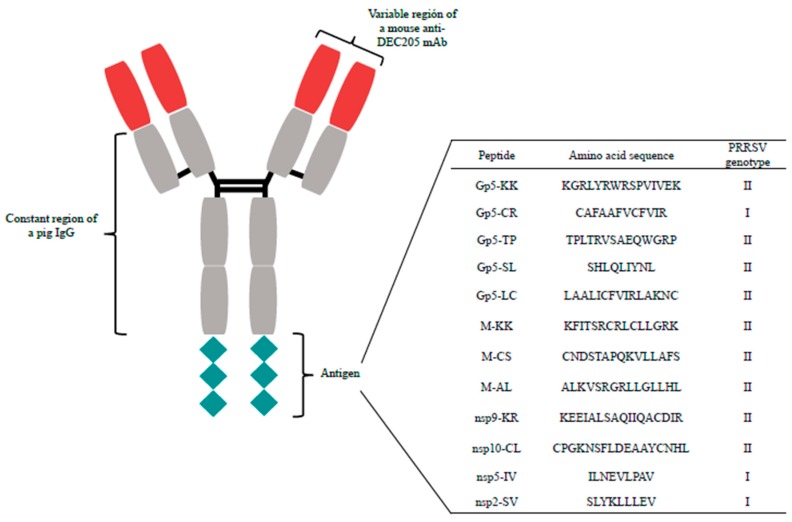
Graphic representation of the recombinant antigenized chimeric mouse x pig antibody utilized. In blue: variable region of a mouse anti-DEC205 monoclonal antibody. In gray: constant region of a pig IgG. The antigenized part consists of a chimeric protein based on selected immunogenic peptides from PRRSV (red).

**Figure 2 vaccines-07-00043-f002:**
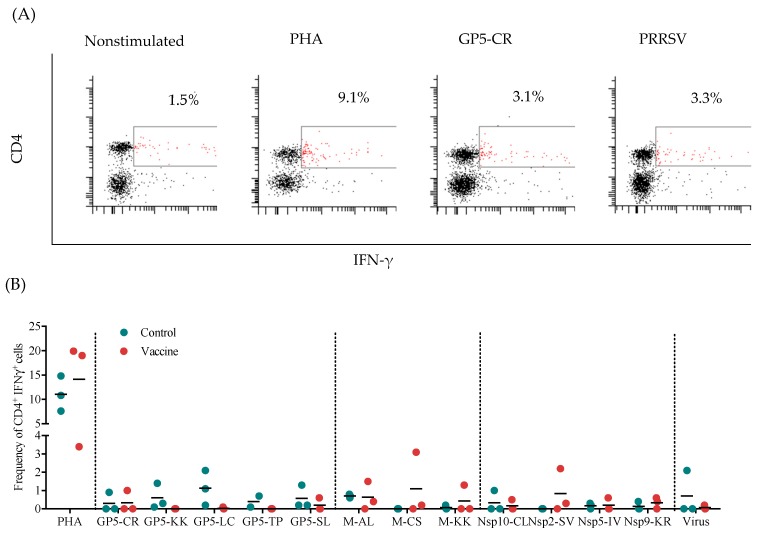
Analysis of CD4^+^IFN-γ-secreting cells. (**A**) Representative experiment with PBMCs stimulated with individual porcine reproductive and respiratory syndrome virus (PRRSV) peptides, the whole virus, phytohaemagglutinin (PHA) (positive control) or medium along (unstimulated control). For extracellular labeling, an anti-CD4 antibody conjugated with fluorescein isothiocyanate (FITC) was used, and for intracellular staining, an anti-pig IFN-γ antibody conjugated with phycoerythrin (PE) was used. The selected population corresponded to CD4^+^IFN-γ^+^ cells. (**B**) Frequency of CD4^+^ IFN-γ-secreting cells in immunized pigs. The results were obtained by subtracting the percentage of CD4^+^IFN-γ^+^ cells in the unstimulated cell group from the percentage of double-positive cells in the experimental group.

**Figure 3 vaccines-07-00043-f003:**
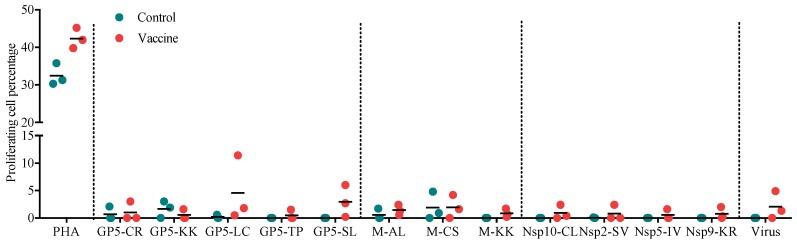
Proliferation of PBMCs from immunized pigs in response to structural and nonstructural peptides and PRRSV. The results were obtained by subtracting the percentage of proliferating cells observed with the unstimulated cells from the percentage observed with the stimulated cells.

**Figure 4 vaccines-07-00043-f004:**
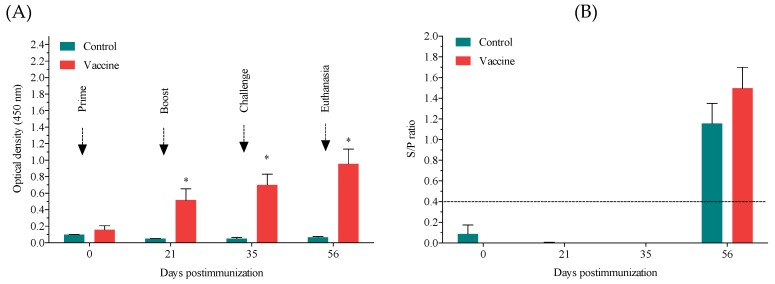
Antibody response to the anti-PRRSV antibody. (**A**) Evaluation of antibodies against the anti-PRRSV antibody at the beginning of the experiment (Day 0), at the time of the booster immunization (Day 21), at the time of the challenge (Day 35), and at the end of the experiment (Day 56) by ELISA. Gray bars represent the control group, and blue bars represent the vaccine group. Bars represent the mean ± standard error (SE) of three animals. (**B**) Antibody response evaluated by a commercial kit from IDEXX to confirm seroconversion in the challenged pigs.

**Figure 5 vaccines-07-00043-f005:**
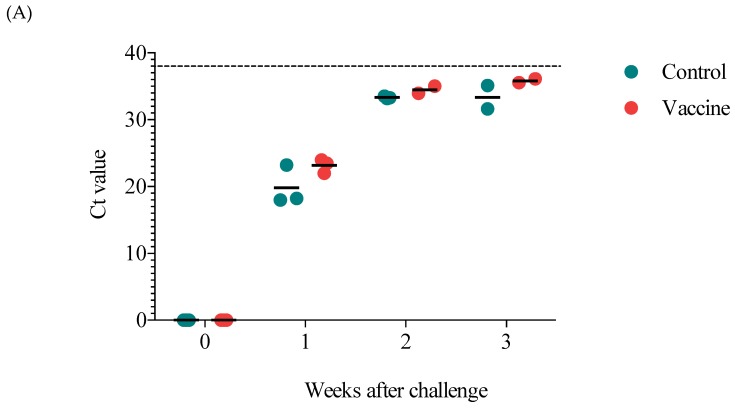
Viremia and the viral load in the lungs and tonsils of challenged pigs. (**A**) Viremia evaluated on the day of the challenge and every week thereafter until the end of the experiment (three weeks postchallenge). The viral load in the lungs (**B**) and tonsils (**C**) at the end of the experiment. The results are expressed as cycle threshold (*Ct*) values, and the dotted line represents the cut-off for qPCR. Each point represents the *Ct* value of one pig.

**Figure 6 vaccines-07-00043-f006:**
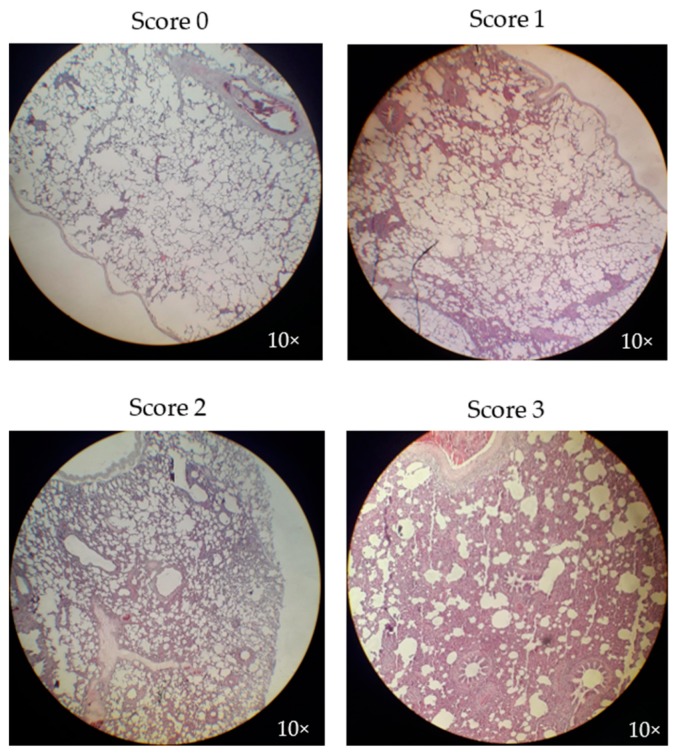
Microscopic lung lesion evaluation of immunized pigs at the end of the experiment. The scoring system utilized to evaluate the severity of the lung lesions found in pigs was as follows: 0—no significant lesions; 1—mild significant lesions; 2—moderate significant lesions; and 3—severe significant lesions. Tissue samples were magnified at 10×.

**Table 1 vaccines-07-00043-t001:** Microscopic lung lesion scores for the groups.

Microscopic Lung Lesions Score	Number of Pigs with Each Score	
	Control Group	Vaccine Group
0	1/3	1/3
1	1/3	0/3
2	1/3	2/3
3	0/3	0/3

The utilized scoring system evaluated the severity of lung lesions in the following manner: 0—no significant lesions; 1—mild significant lesions; 2—moderate significant lesions; and 3—severe significant lesions (see Figure 3).

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
