# Peer review of "Evaluation of a Recombinant Mouse X Pig Chimeric Anti-Porcine DEC205 Antibody Fused with Structural and Nonstructural Peptides of PRRS Virus"

_vaccines, 2019, doi:10.3390/vaccines7020043_

Round 1

Reviewer 1 Report

authors have addressed the issues and the paper is ready to be accepted

Reviewer 2 Report

In this version, the authors addressed most of the concerns from this reviewer. Although the statistical analysis is not perfect, it is good to publish.

Reviewer 3 Report

The manuscript has improved,  in particular to tone down the significance of their work and to cut down the unnecessary literature and discussion.

But what is done is done.  Without further experiments, this is the limit that authors can present.

This manuscript is a resubmission of an earlier submission. The following is a list of the peer review reports and author responses from that submission.

Round 1

Reviewer 1 Report

Dec205 fusion is important and the concept to use the fused antibody to deliver T- and B-cell epitopes are novel and innovative. The manuscript is well-written and carefully balance the pros and cons. It's not oversold. I think minor revision to provide a better discussion of how best to improve would be great!

Reviewer 2 Report

This study investigated the efficacy of a recombinant mouse x pig chimeric antibody against PRRSV infection in pigs. Dendritic cell-targeting strategy was used in this this study through construction of a panel of chimeric mouse x pig antibodies loading different antigens of PRRSV. Although the efficacy of the novel vaccines generated in this research was not as good as expected according to the data present in this manuscript, this study, together with other researches on dendritic cell targeting technique, provided significant references to explore novel PRRSV vaccines. Major concerns from this reviewer are as below.

1.      In the manuscript, the authors used a commercial PRRSV ELISA kit, herdchek PPRS Antibody Test Kit, IDEXX Laboratories. What’s the purpose of this ELISA here? This kit only detect antibody against N protein of PRRSV, but the vaccines used in this study never loaded any N protein antigen. And, there is no data or any description about the result of this test in result part.

2.      The statistics analysis used in some data was not correct. According to the fig 2/3/5, looks that the value 0 was taken into account. This is not right.

3.      The manuscript need lots of editorial work. Some figures needed to be rearranged, for example fig 2/5, and many typo were found. The figure 3 in page 6 should be figure 2.

Reviewer 3 Report

This manuscript reports a study on the efficacy of a chimeric mouse/porcine monoclonal antibody against a cell-surface molecule receptor molecule on dendritic cells (described as DEC205): immunogenic structural and non-structural peptides from porcine respiratory and reproductive virus (PRRSV) are attached at the Fc part of the chimeric antibody. The rationale for this work is the persistent need for an effective vaccine against this virus in the commercial swine industry. The DEC205 receptor has been used in other studies to increase efficiency in antigen presentation. The aim of the work was to assess the immunogenicity and the protection against PRRSV upon intradermal injection of the monoclonal antibody. The structure of this product, i.e., the variable regions of the light and heavy chain from the mouse monoclonal antibody, the constant regions and the Fc part from a pig antibody, and the attachment of PRRSV antigens at the Fc location, is well explained. The in vivo work was performed in 6 young pigs from a farm that was documented to be free of PRRSV: three of these received the antibody and three controls received a physiological saline injection, with a boost three weeks after the first injection. A PRRSV challenge was performed two weeks after the booster injection, and the animals were terminated three weeks later. The immunized pigs manifested the formation of IgG class anti-PRRSV antibodies in the circulation. Interferon-gamma positive CD4-positive cells in peripheral blood mononuclear cells upon stimulation with individual peptides or virus, performed two/three weeks after the challenge, did not reveal clear differences between the two groups. Histology for PRRSV-associated lesions in lung samples. The same was observed for a proliferative response upon stimulation. Samples at necropsy were assessed for the presence of viral RNA by qRT-PCR in samples from serum, tonsil and lung. The outcomes in serum are described as viremia and those in tissue as viral load. There were no clear differences between the two groups. The same applies for microscopic lesions in lung histology. It is concluded that following the immunization and booster injection with the modified antibody only immunogenicity of the PRRSV peptides could be demonstrated, but no protection against the virus.

This is an interesting study on an important topic in commercial swine industry. But there are a number of concerns.

Major concerns

·         The model to assess protection might not be suitable. Since there were only two animals in the control group presenting lesions in lung histology, and since in one of these two the lesions were slight and in the second moderate, this outcome parameter has (1) variability and (2) too little changes to serve as control. Essentially, all three controls should have at least moderate lesion to enable a comparison with vaccine interference. The same applies to the outcomes regarding viremia and viral load. It might be that the challenge with PRRSV was at a too low dose level, which also explains the absence of clear clinical signs.

·         The targeting to dendritic cells is not documented in this study, only claimed by referencing literature studies. So, the conclusion in the discussion, lines 263-266, might not be valid regarding the targeting of DEC205-positive dendritic cells. The same applies to lines 311-312 in the Conclusions. The rationale for intradermic administration is based on literature data.

·         There is no validation of cellular immune response assays. The internal control using phytohaemagglutinin is fine, but this does not address the response to immunization. This needs to be documented in a separate study. In other words, it is not clear whether the assays used are able to detect a cell-mediated immune reactivity towards PRRSV or PRRSV vaccines. Finally, it can be questioned whether the performance of the respective in vitro assays only at two/three weeks after challenge, i.e., seven/eight weeks after the first immunization and four/five weeks after the booster injection makes sense to demonstrate a status of cellular immunity.

·         The number of animals is quite low (n=3 per group). However, there are differences between  the three animals in the immunized group that merit attention in a case-by-case evaluation.. One case after immunization showed a higher frequency of interferon-gamma positive CD4-positive cells regarding M-AL, M-CS, Nsp-SV: regarding the proliferation of blood mononuclear cells upon stimulation also one case in the immunized group showed stimulation with some peptides, and also with virus. Is this the same animal in these assay, or is the apparent higher response intrinsic to the variability in these assays?

Minor points

·         Abstract: it is advised to clarify DEC205 as a receptor on dendritic cells.

·         Introduction. It is stated that structural and non-structural peptides were selected from previous work. The rationale underlying this selection needs to be explained.

·         The challenge of PRRSV is not clarified regarding the route of administration.

·         Antibody assays. For one assay it seems clear that the outcome regards IgG-class antibody, for the other assay this is not clear.

·         There are two figures 3 in the manuscript. Both figures do not show the control upon PHA mitogen stimulation.

·         Figure 4. It is recommended to present antibody responses for individual animals, just like in both figures 3. It seems that only the result for the assay developed in the authors’ laboratory is presented, and not the results using the commercial kit. This needs explanation. Were the data is both assays the same?

·         Figure 6. The histology needs to be documented at higher magnification. Also, size bars need to be included.

·         In line 280 it is concluded that “Further studies are needed to find better routes, antigens and/or doses to stimulate IFNÎł production.” This is essentially a killing statement regarding the value of this study. To give this statement value it needs to be expanded with statements underscoring what is the possible reason for the finding of only IgG-antibodies and the value of such antibodies, and the outline what need to be addressed in further studies in view of the limitations of the present study.

·         The statement in the discussion that this study did not include neutralizing antibodies is a clear limitation of the study, and needs to be explained in more detail

·         The conclusion in lines 285-286 that “our results clearly show an enhanced humoral response that was not observed in other studies where bigger antigenic proteins were used in DCs targeting” needs evidence in showing actual data.

Reviewer 4 Report

In the abstract, the authors contradicts to themselves.  In line 19 they said"limited protection", and yet in line 23 they said "better level of protection", which one?

Line 73: how the animal was challenged, intranasal or per oral or intramuscular ???

The number of experimental pigs: n= 3 in each group is minimal for statistical analysis.  The number is important when the results are not so significant, although we know pig experiment is not easy.

There is no figure 2.

To me, the only significant results are in Fig. 3A flow histogram and Fig. 4.

In Figs. 5-6, Table 1 there are no difference between control and vaccine groups, again the number of experimental animals in each group matters when the results are vague, and I know you the authors try hard to present.

line 138, reference Silva-Campa 2012 is not listed.

In several places in the text, Subramaniam were mentioned but in the reference listing you have three Subranmaniam, specifies which one is which in the text.

In reference listing, some journal names are abbreviated, while others are not.  Some page number are fully cited, while some are shortened.  This is the basics of publishing.